# Ameliorating Phosphonic-Based Nonflammable Electrolytes Towards Safe and Stable Lithium Metal Batteries

**DOI:** 10.3390/molecules28104106

**Published:** 2023-05-15

**Authors:** Sha Fu, Xuanzhi Xie, Xiaoyi Huangyang, Longxi Yang, Xianxiang Zeng, Qiang Ma, Xiongwei Wu, Mingtao Xiao, Yuping Wu

**Affiliations:** 1School of Chemistry and Materials Science, Hunan Agricultural University, Changsha 410128, China; fusha456@163.com (S.F.); 13787333800@163.com (X.H.); yanglongxi@stu.hunau.edu.cn (L.Y.); xxzeng@hunau.edu.cn (X.Z.); 2School of Mechanical and Electrical Engineering, Hunan Agricultural University, Changsha 410128, China; xiexuanzi1998@163.com; 3Henan International Joint lof Rare Earth Composite Materials, Henan University of Engineering, College of Materials Engineering, Zhengzhou 451191, China; 4School of Energy and Environment, Southeast University, Nanjing 210096, China; wuyp@fudan.edu.cn

**Keywords:** lithium metal batteries, metallic lithium anode, electrode–electrolyte interface, interface compatibility, nonflammable electrolytes

## Abstract

High-energy-density lithium metal batteries with high safety and stability are urgently needed. Designing the novel nonflammable electrolytes possessing superior interface compatibility and stability is critical to achieve the stable cycling of battery. Herein, the functional additive dimethyl allyl-phosphate and fluoroethylene carbonate were introduced to triethyl phosphate electrolytes to stabilize the deposition of metallic lithium and accommodate the electrode–electrolyte interface. In comparison with traditional carbonate electrolyte, the designed electrolyte shows high thermostability and inflaming retarding characteristics. Meanwhile, the Li||Li symmetrical batteries with designed phosphonic-based electrolytes exhibit a superior cycling stability of 700 h at the condition of 0.2 mA cm^−2^, 0.2 mAh cm^−2^. Additionally, the smooth- and dense-deposited morphology was observed on an cycled Li anode surface, demonstrating that the designed electrolytes show better interface compatibility with metallic lithium anodes. The Li||LiNi_0.8_Co_0.1_Mn_0.1_O_2_ and Li||LiNi_0.6_Co_0.2_Mn_0.2_O_2_ batteries paired with phosphonic-based electrolytes show better cycling stability after 200 and 450 cycles at the rate of 0.2 C, respectively. Our work provides a new way to ameliorate nonflammable electrolytes in advanced energy storage systems.

## 1. Introduction

Unoptimized energy structures, such as the excessive utilization of fossil energy, will cause environmental pollution and greenhouse effect problems. Under this background, the development of renewable clean energy, including wind, solar and electrochemical storage technology, is of great significance to the sustainable application of energy. Among them are rechargeable lithium metal batteries (LMBs); their high-energy-density has drawn much attention around the world [1,2,3,4,5]. In recent years, LMBs have been regarded as the ideal choice to replace traditional lithium-ion batteries in the stationary energy storage field and electric vehicles [6,7,8,9,10]; however, accompanied by the application of highly active metallic Li anodes, safety concerns for LMBs also increased. The metallic Li anode easily reacts with organic liquid electrolytes, causing the uneven deposition of metallic Li or the formation of Li dendrite, which will deteriorate the electrochemical performance of LMBs. Meanwhile, the Li dendrite will penetrate the membrane, leading to the short circuiting of LMBs and even to thermal runaway [11,12,13,14,15,16], such as the spontaneous combustion of smartphones and electric vehicles [17,18,19,20]. These safety failures largely originated from commercial organic electrolytes, which have low flash points and are prone to combustion [21,22,23,24,25]. Hence, developing non-flammable electrolyte is one of the most effective measures to reduce the flammability of electrolytes and improve the safety of LMBs.

Up to now, much effort has been devoted to the development of nonflammable electrolytes, such as ionic liquid [26,27,28], fluorine-containing solvent [29,30,31,32], organic phosphate ester solvent [33,34,35,36,37] and solid-state electrolytes (SSEs) [38,39,40,41,42]. Although SSEs are considered as alternative electrolytes for the development of LMBs, the low ionic conductivity and inferior interface compatibility limit their further large-scale application in high-energy-density LMBs [43,44,45,46,47]. Due to the superior flame retardation and cost effectiveness, organic phosphate became one of many promising solvents in nonflammable electrolytes [48,49,50]; however, most phosphate ester flame retardants usually show poor compatibility with lithium metal anodes. Flame retardant molecules will be stripping/insetting together with solvated lithium ions, leading to electrode structural damage and affecting the battery’s electrochemical performance. Some studies have shown that salts with a (fluoro sulfonyl)imide group (FSI^-^) preferentially coordinate with organic phosphate molecules, then they add high reduction potential Li^+^-coordinated additives, e.g., fluoroethylene carbonate (FEC) and vinylene carbonate (VC). Under the synergistic action of these two kinds of substances, uniform SEI is formed to improve the electrochemical performance of the battery [51,52,53,54]. It is necessary to ameliorate the structure and functional constituent to improve the compatibility of organic phosphate-based electrolyte with metallic Li anodes and Ni-rich cathodes. Dimethyl allyl-phosphonate (DMAP) shows a flame-retardant effect and superior compatibility with layered Ni-rich cathode; however, the inferior complex behavior with lithium salt impeded their further usage.

Herein, the functional additives FEC and DMAP were introduced to triethyl phosphate electrolytes to stabilize the deposition of metallic lithium and ameliorate the electrode–electrolyte interface environment. Meanwhile, the FEC will be reduced at the metallic Li anode side to build the F-rich solid electrolyte interface (SEI). Furthermore, the dimethyl allyl-phosphate could be oxidated at the Ni-rich cathode surface to guarantee the stability of the cathode and electrolyte interface. In virtue of the designed flame-retardant electrolytes, the Li||Li symmetrical batteries exhibit an excellent cycling stability of 700 h at the condition of 0.2 mA cm^−2^, 0.2 mAh cm^−2^. In addition, the smooth and dense Li anode surface was observed on cycled Li symmetrical batteries, demonstrating that the designed electrolytes show better interface compatibility with the metallic lithium anode. The Li||LiNi_0.8_Co_0.1_Mn_0.1_O_2_ and Li||LiNi_0.6_Co_0.2_Mn_0.2_O_2_ batteries paired with phosphonic-based electrolytes exhibit superior cycling stability after 200 and 450 cycles at the rate of 0.2 C, respectively. Our work sheds light on a new strategy to optimize nonflammable electrolytes in advanced energy storage systems.

## 2. Results and Discussion

The lithium dendrite formation is notorious during the charge and discharge of LMBs, which severely limit the utilization efficiency and capacity of metallic lithium anodes. The fundamental reason for this is that the highly active lithium will directly react with organic electrolyte to unevenly form a solid electrolyte interface. The lithium dendrite will puncture the separator, resulting in short circuits inside the batteries and even fire and explosion. The usage of flame-retardant electrolytes is an effective way to guarantee the safety of LMBs during cycling; however, most flame-retardant electrolytes suffer from inferior interface compatibility with metallic lithium anodes and layered Ni-rich cathodes. Based on the aforementioned problems, the functional additives FEC and DMAP were introduced to triethyl phosphate electrolytes. As shown in Figure 1a,b, the inflaming retarding capacity of DMAP electrolytes was evaluated. DMAP electrolytes still do not burn when ignited with an alcohol spray gun, whereas conventional organic carbonic ester-based electrolytes are known to burn extremely easily. Compared to the traditional organic carbonic ester-based electrolytes, the designed electrolytes show excellent flame-retardant performance. Additionally, the chemical component and composition of organic phosphate electrolytes was analyzed by FTIR, and the result are shown in Figure 1c. The peaks located at 920 cm^−1^, 1020 cm^−1^, 1200 cm^−1^, 1640 cm^−1^, 1400 cm^−1^ belong to the P-O, P-O-C, P=O, C=C and C-F functional groups, which is in accord with the structure of FEC and DMA9P additives. As shown in Figure 1d, commercial electrolytes evaporate and the quality drops sharply along with the rising temperature. It is interesting that the organic phosphate electrolytes exhibit superior thermal stability at the temperature window ranging from 30 to 135 °C.

The electrochemical performance of the designed electrolyte was further evaluated. Considering the electrolyte paired with the high-voltage cathodes, the voltage window of the DMAP electrolyte was tested in the coin battery. The result shows that the stable voltage could reach up to 4.7 V (Figure 2a), which can meet the requirements of the high voltage and capacity of the Ni-rich cathode and LiNi_0.8_Mn_0.1_Co_0.1_O_2_. Furthermore, the variable temperature impedance was measured by assembling SS||SS symmetrical battery with a test temperature ranging from 30 to 80 ℃ (Figure 2b). The fitting result is shown in Figure 2c, and the activation energy is 0.23 eV, indicating that the DMAP electrolyte possesses fast ion transfer capacity.

The Li||Li symmetrical batteries with DMAP electrolytes and commercial electrolytes were assembled. In comparison with the batteries paired with commercial electrolytes, the symmetrical batteries with DMAP electrolytes show an outstanding cycling stability for 700 h at the condition of 0.2 mA cm^−2^, 0.2 mAh cm^−2^ (Figure 3a). The morphology of the lithium anode for cycled symmetrical batteries was observed by SEM. The needle-like lithium dendrites were observed (Figure 3b,c). On the contrary, the homogeneous and dense Li surface was formed in the symmetrical batteries with DMAP electrolytes, indicating that the designed electrolytes possess an excellent interface compatibility with metallic lithium anode (Figure 3d,e).

Furthermore, the Li||LiNi_0.8_Co_0.1_Mn_0.1_O_2_ and Li||LiNi_0.6_Co_0.2_Mn_0.2_O_2_ batteries with DMAP electrolytes were assembled to their evaluate electrochemical performance. As shown in Figure 4a,e, the Li|DMAPs|LiNi_0.8_Co_0.1_Mn_0.1_O_2_ battery exhibits outstanding stability with 85% capacity retention after 200 cycles. In addition, the Li|DMAPs|LiNi_0.6_ Mn_0.2_Co_0.2_O_2_ battery also shows high-capacity retention of 92.5% after 450 cycles at 0.2 C (1 C = 180 mA h g^−1^), shown in Figure 4d, which demonstrated that the designed electrolytes show better compatibility with the cathode. Furthermore, The EIS result of the battery was tested before and after the cycling. It was found that the impedance decreased after cycling, which was related to the formation of SEI or CEI (Figure 4b). In addition, the reversibility of Li|DMAPs|LiNi_0.8_Mn_0.1_Co_0.1_O_2_ batteries was confirmed by cyclic voltammetry. As shown in Figure 4c, the curves of the second and the third are basically consistent after the first activation of the Li|DMAPs|LiNi_0.8_Co_0.1_Mn_0.1_O_2_ battery, demonstrating that the introduction of the DMAP electrolytes is conducive to enhancing the reversibility of the batteries.

The chemical composition of the metallic Li surface after cycling in Li||NCM622 battery with DMAP electrolyte was further analyzed by XPS (Figure 5a–e). Combined with P 2p, C 1s, F 1s and N 1s spectra (Figure 5b–e), some functional chemical bonds were detected, such as P-O, LiF, etc., which is conducive to ameliorate the electrode and electrolytes interface. It is worth noting that the inorganic component LiF was observed on the metallic Li surface, demonstrating that the FEC and LiTFSI participate in the formation of SEI, which is beneficial to the stable deposition of metallic lithium. In addition, the nitrogen-containing chemical bonds in Figure 4f come from the additive LiNO_3_. LiNO_3_ binds to the solvent or active substance in the electrolyte, and Li-N can reduce the interface impedance and, thus, enhance the migration of lithium ions.

## 3. Materials and Methods

### 3.1. Materials

Bis(trifluoro methane sulfonyl)imide lithium salt (LiTFSI, 99%), lithium nitrate (LiNO_3_, AR, 99%), triethyl phosphate (TEP, 99+%), vinylene carbonate (VC, <2%BHT, 98%), dimethyl allyl-phosphonate (tech-85%), poly(1,1-difluotoethylene) (PVDF, average Mw: 800,000), conductive carbon black (Super P) and LiNi_0.8_Mn_0.1_Co_0.1_O_2_ (NCM811) were purchased from Guangdong Canrd New Energy Technology Co., Ltd. 1-Methyl-2-pyrrolidinone (NMP, AR, 99%) and 4-Fluoro-1,3-dioxolan-2-one (FEC, 99%) were purchased from Innochem (Beijing, China).

### 3.2. Preparation and Characterization of Phosphonate-Based Electrolyte

The lithium salt LiTFSI (1 × 10^−3^ mol) is dissolved in 1 mL TEP solvent. The prepared solution is stirred until it is a colorless transparent solution. Then, the 0.5 w% lithium nitrate and 1 w% FEC are added to the colorless transparent solution, stirring for 10 h. After the stirring is completed, 3 w% organic phosphate dimethyl allyl-phosphonate is added into the above solution.

Fourier transform infrared spectroscopy (FT-IR, Alpha) was collected in the range from 400 to 4000 cm^−1^. X-ray photoelectron spectroscopy (XPS, PHI Versa Probe 4, ULVAC-PHI, Kanagawa, Japan) was conducted with 300 W Al Kα radiation. All peaks would be calibrated with C 1s peak binding energy at 284.8 eV for adventitious carbon. The thermogravimetric analysis (TGA) and differential scanning calorimeter were performed using Metter Toledo instrument over a temperature range from 30 to 400 °C under a nitrogen atmosphere with a heating rate of 5 °C min^−1^. Scanning electron microscopy (SEM, JSM-7610FPlus, JEOL, Tokyo, Japan) was performed at an acceleration voltage of 10 kV. Energy dispersive spectroscopy (EDS) mapping images were collected by Oxford (ULTIM MAX 40, Oxford Instruments, Abingdon, UK). Raman spectrometer was collected in the Raman shift range from 25 to 3500 cm^−1^ and 532 nm laser excitation was used. Ball mill was performed using Miqi YXQW-planetary muller.

### 3.3. Electrochemical Measurements

The cathode slurry was prepared by blending LiNi_0.8_Co_0.1_Mn_0.1_O_2_ (NCM811) or LiNi_0.6_Co_0.2_Mn_0.2_O_2_ (NCM622), super P and poly(vinylidene fluoride) with a mass ratio of 8:1:1. The average loading of active material in each pellet was about 2.0–3.0 mg cm^−2^. The electrochemical performances were tested by assembling 2032-type coin cells with NCM811 or NCM622 as the cathode, metallic Li as the anode and using designed electrolytes, respectively. All coin cells were assembled in an Ar-filled glovebox (the oxygen and water concentration maintained below 0.1 ppm). The galvanostatic charge/discharge tests with a voltage range from 2.8 to 4.3 V were performed on a battery test system (NEWARE BTS, Shenzhen, China) at room temperature. Cyclic voltammetry (CV) was tested with a scan speed of 0.1 mV s^−1^ using an electrochemical workstation (Chenhua 760D, Shanghai Chenhua Instrument Co., LTD., Shanghai, China) with the voltage window from 2.8 to 4.3 V. Electrochemical impedance spectroscopy was carried out with a voltage amplitude of 10 mV at a frequency ranging from 0.01 Hz to 100 kHz. The electrolytes were characterized by FTIR, XPS and Raman. The cells were disassembled after the charge/discharge process. The electrodes were rinsed with solvent (TEP) and dried for characterization.

The phosphonate-based cell’s Coulombe efficiency (CE) was calculated using Equation (1):(1)$$\mathrm{CE}\;=\;\frac{\mathrm{Discharge}\;\mathrm{capacity}}{\mathrm{Charge}\;\mathrm{capacity}}$$

The charge–discharge rate of Li-NCM811 (1 C = 200 mAh g^−1^) coin batteries was set to 0.1 C, and the charge–discharge current density of the Li-NCM811 coin batteries was calculated according to Equation (2):(2)$$\mathrm{Current}\;\mathrm{density}\;=\;\frac{C\;-\;\mathrm{rate}\times \mathrm{ominal}\;\mathrm{specific}\;\mathrm{capacity}}{\mathrm{cathode}\;\mathrm{area}}$$

The activation energy was obtained from EIS spectroscopy using a stainless steel cell, with the tested temperature ranging from 30 to 80 °C. The ionic conductivity (σ) of the phosphate electrolyte was calculated according to Equation (3):(3)$$\sigma =A*e^{\frac{-\mathrm{Ea}}{\mathrm{RT}}}$$
where A is the frequency factor, Ea is the activation energy, R is the molar gas constant and T is the absolute temperature.

Equation (3) is commonly used to calculate ionic conductivity, and Equation (3) is closely related to the activation energy data mentioned in the manuscript and the data fitting in Figure 2c.

## 4. Conclusions

In summary, we designed the novel nonflammable electrolytes by introducing the functional additives fluoroethylene carbonate (FEC) and dimethyl allylphosphate to triethyl phosphate electrolytes. The FEC could be reduced at the metallic Li anode side to build F-rich solid electrolyte interface (SEI), mitigating the even deposition of metallic lithium. Moreover, the dimethyl allylphosphate could be oxidated at the Ni-rich cathode surface to guarantee the stability of cathode and electrolyte interface. Benefiting from the DMAP flame-retardant electrolytes, the Li||Li symmetrical batteries exhibited an excellent cycling stability of 700 h. In addition, the smooth and dense Li anode surface was obtained on the cycled Li symmetrical batteries, demonstrating that the DMAP electrolytes show better interface compatibility with metallic lithium anode. The Li||LiNi_0.8_Co_0.1_Mn_0.1_O_2_ and Li||LiNi_0.6_Co_0.2_Mn_0.2_O_2_ batteries paired with phosphonic-based electrolytes exhibited superior cycling stability after 200 and 450 cycles at the rate of 0.2 C, respectively. Our work provides a new strategy to optimize nonflammable electrolytes in high energy density lithium metal batteries.

## Figures and Tables

**Figure 1 molecules-28-04106-f001:**
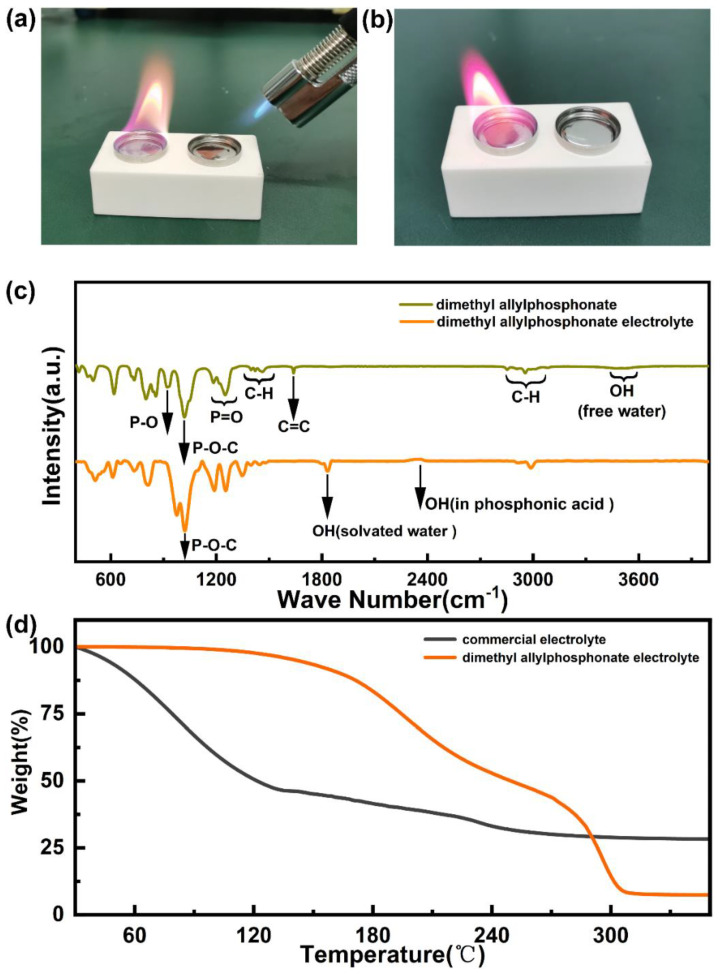
The thermal stability comparison of commercial electrolyte and DMAP electrolyte (**a**,**b**); the FTIR measurement result of DMAP and DMAP electrolytes (**c**); (**d**) TGA test of the commercial electrolyte and DMAP electrolyte.

**Figure 2 molecules-28-04106-f002:**
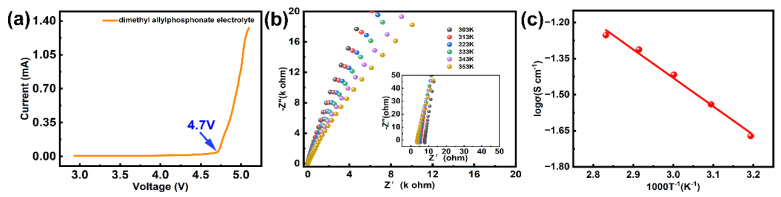
(**a**) The electrochemical window of DMAP-based electrolytes. The electrochemical window of electrolyte generally favors steel symmetric battery. The test starts with the open circuit voltage of the battery and 0.1 mV s^−1^ sweep speed is selected. (**b**,**c**). (**c**) is the linear fitting result of (**b**) the first five different temperature AC impedance data. The variable temperature impedance of the DMAP-based electrolytes and the corresponding fitting result. It is recommended to test the battery at different temperatures, from high to low, and to stabilize the battery at each test temperature for half an hour before testing.

**Figure 3 molecules-28-04106-f003:**
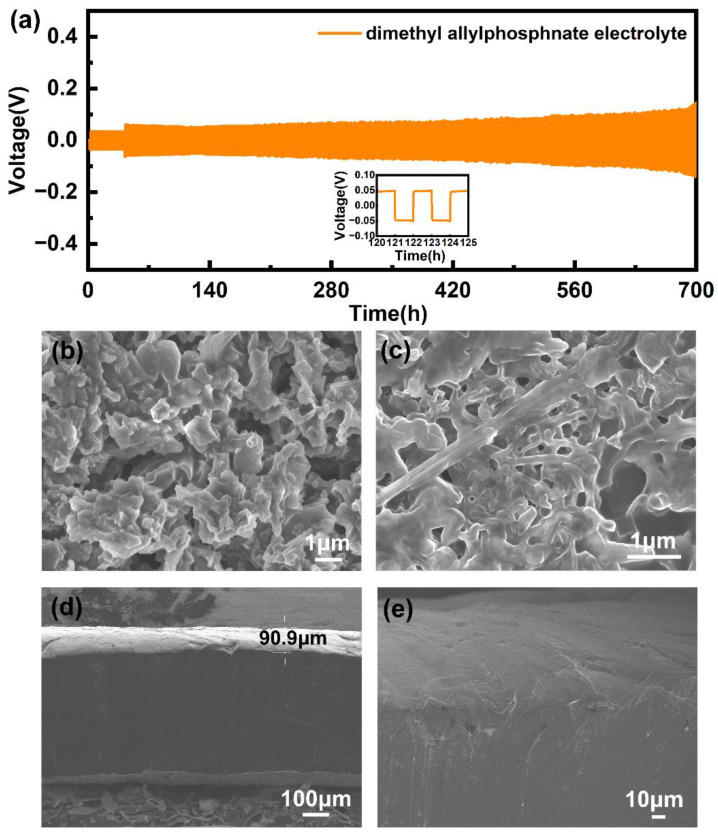
(**a**) The cycling performance of Li||Li symmetrical battery with DMAP electrolytes; the SEM morphology for cycled metallic Li anode in (**b**,**c**) commercial electrolytes and (**d**,**e**) DMAP electrolytes (cross section, 90.9 μm can be considered as the thickness of SEI layer formed by lithium deposition).

**Figure 4 molecules-28-04106-f004:**
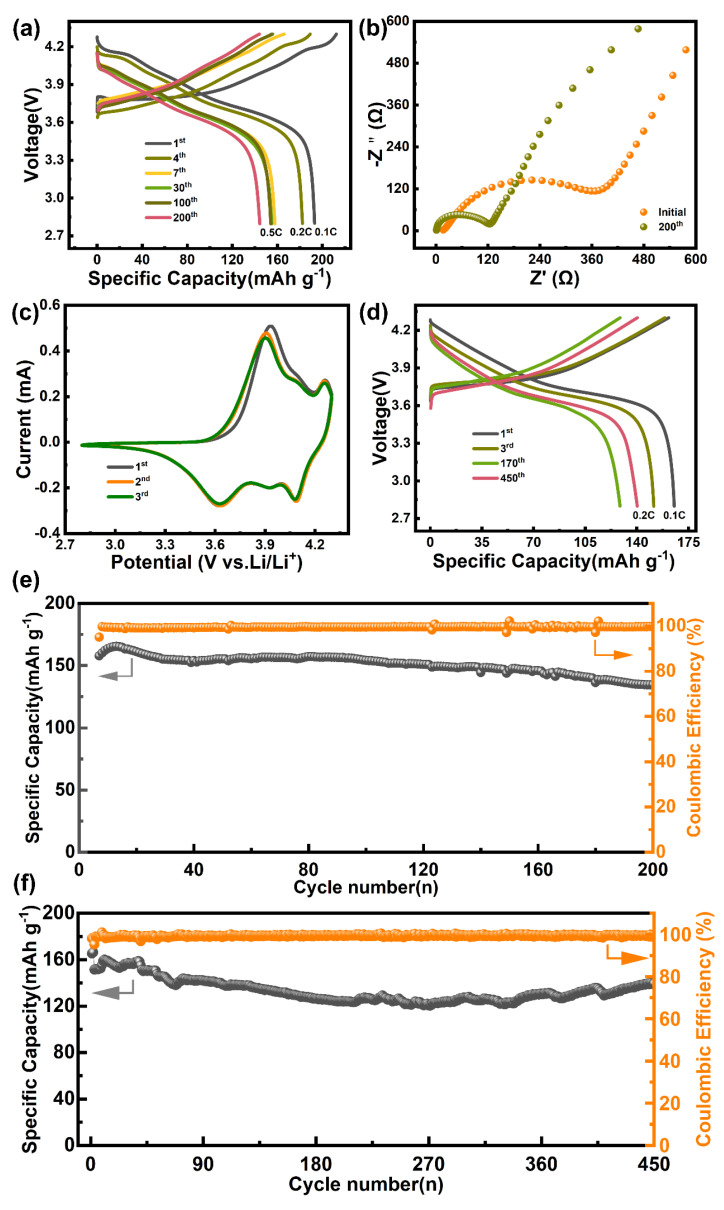
Electrochemical performance of the whole battery. (**a**) the discharge capacity comparison of Li|DMAPs|LiNi_0.8_Co_0.1_Mn_0.1_O_2_ battery; (**b**) the EIS result of Li|DMAPs|LiNi_0.8_Co_0.1_Mn_0.1_O_2_ battery before and after cycling; (**c**) the CV curves of Li|DMAPs|LiNi_0.8_Co_0.1_Mn_0.1_O_2_ battery; (**d**) the discharge capacity comparison of Li|DMAPs|LiNi_0.6_Co_0.2_Mn_0.2_O_2_ battery and the cycling stability of Li||NCM811 (**e**); and (**f**) Li||NCM622.

**Figure 5 molecules-28-04106-f005:**
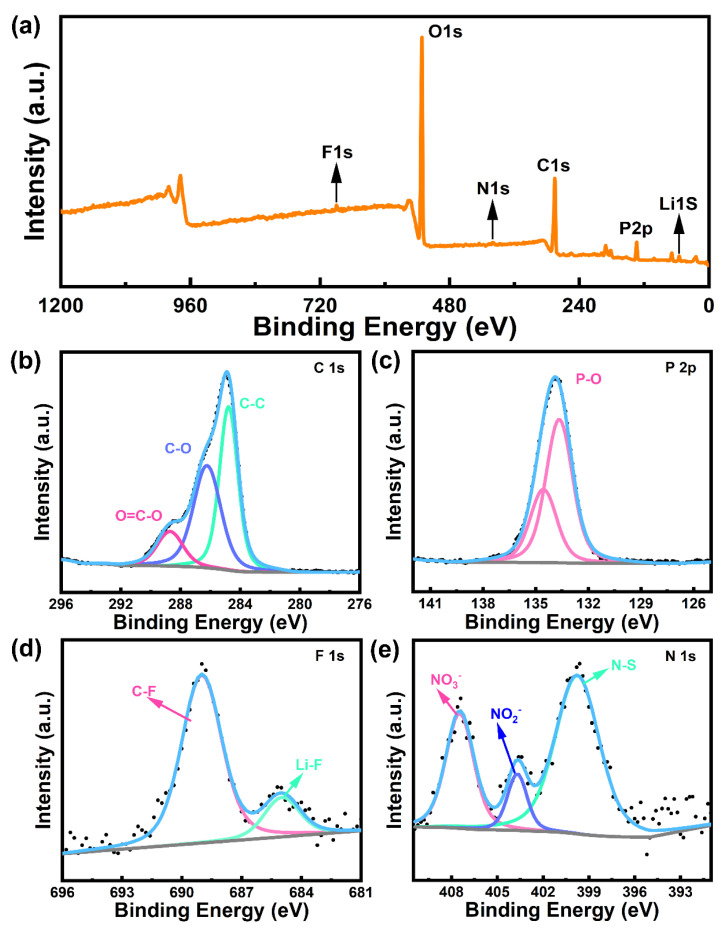
(**a**) the full spectrum of the XPS characterization of Li-NCM622 batteries after 450 cycles in the electrolyte containing 3% dimethyl allyl-phosphonate; the surface information for cycled metallic Li (**b**) C1s, (**c**) P2p, (**d**) F1s and (**e**) N1s; The XPS fitting literature is based on Refs. [34,35]. The dots are the results of XPS data, and the lines are the fitting results of the dots.

## Data Availability

Not applicable.

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
