# Peer review of "Ameliorating Phosphonic-Based Nonflammable Electrolytes Towards Safe and Stable Lithium Metal Batteries"

_molecules, 2023, doi:10.3390/molecules28104106_

Round 1

Reviewer 1 Report

This research reports an interesting work of designing a nonflammable electrolytes for NCM-based lithium ion batteries with both high safety and high capacity, which is a hotspot of vital importance to the area of lithium ion batteries. The results are well characterized and the cell performances can be comparable with commercial electrolyte-based cells. However, some questions should be clarified as following:

1. What is the addictive amount of dimethyl allylphosphonate and FEC, as also as LiNO3, in the TEP-LiTFSI electrolyte?

2. What is the role of VC, which is listed in  Section 3.1? And the information of FEC should be given in this section.

3. The result of ionic conductivity seems to be much higher than that of traditional carbonate-based electrolyte with LiTFSI, what is the probable reason?

4. The authors should confirm the calculation equation of ionic conductivity (equation 3), or to clarify the use of equation 3.

5. There are some mistakes in the manuscript, e.g. the spell of LiNixCoyMnzO2, the concentration of LiTFSI-TEP (1 mol in 1 mL), and some others.

Author Response

For Reviewer #1:

This research reports an interesting work of designing a nonflammable electrolytes for NCM-based lithium ion batteries with both high safety and high capacity, which is a hotspot of vital importance to the area of lithium ion batteries. The results are well characterized and the cell performances can be comparable with commercial electrolyte-based cells. However, some questions should be clarified as following:

Response: Thanks for your positive evaluation about our work.

  1. What is the addictive amount of dimethyl allyl-phosphonate and FEC, as also as LiNO3, in the TEP-LiTFSI electrolyte?

Response 1: Thanks for the comments. The additive is added in small amounts, so the author added 3w% dimethyl allyl-phosphonate, 1w% FEC and 0.5% LiNO3 in the TEP-LiTFSI electrolyte. Correspondingly, we have made corresponding modification in the revised manuscript (see Page 8, lines 2-4).

  1. What is the role of VC, which is listed in Section 3.1? And the information of FEC should be given in this section?

Response 2: Thanks for the comments. First, the NCM cathodic electrode is prone to overcharging in the course of the actual experiment. As the reviewer mentioned VC which can be used as overcharge protection additive to prolong the cycle life of lithium metal battery. Then, the information about FEC, we will add it in this section 3.1. To address the reviewer’s concern, we have added the information of FEC in the revised manuscript (see Page 7, lines16).

  1. The result of ionic conductivity seems to be much higher than that of traditional carbonate-based electrolyte with LiTFSI, what is the probable reason?

Response 3: Thanks for the comments. To address the reviewer’s concern, we have reviewed the relevant papers. Secondly, according to previous researches (Chinese Journal of Power Sources .09(2007):687-689, Science Bulletin 2022, 67, 1581.), it can be summarized as follows. Compared with traditional carbonate-based electrolyte, TEP solvent had larger molecular weight, higher melting point and lower dielectric constant, which increased the interaction force between the ions of electrolyte and also increased the freezing point of the electrolyte, inhibited the dissociation of lithium salt LiTFSI, which greatly affected the ionic conductivity of TEP electrolyte.

  1. The authors should confirm the calculation equation of ionic conductivity (equation 3), or to clarify the use of equation 3.

Response 4: Thanks for the comments. The reviewer raises a good issue, so we will give an explanation for that equation 3 is commonly used to calculate ionic conductivity, and equation 3 is closely related to the activation energy data mentioned in the manuscript and the data fitting in Figure 2 (c), so the author wrote this general equation in Section 3.3. We have added corresponding explanations in the revised manuscript (see Page 8, lines 46-48).

  1. There are some mistakes in the manuscript, e. g. the spell of LiNixCoyMnzo2, the concentration of LiTFSI-TEP (1 mol in 1 mL), and some others.

Response 5: Thanks for your valuable comments. We have re-checked and corrected these mistakes in the revised manuscript (see Page 3, line 13; see Page 5, line 14; see Page 8, line 1).

Reviewer 2 Report

This manuscript entitled “Ameliorating phosphonic-based nonflammable electrolytes towards safe and stable lithium metal batteries” developed a strategy to handle the interface compatibility and stability faced by nonflammable electrolytes via introducing functional additive dimethyl allylphosphate and fluoroethylene carbonate into triethyl phosphate electrolytes to stabilize the deposition of metallic lithium and accommodate the electrode-electrolyte interface. Dimethyl allylphosphonate shows superior compatibility with layered Ni-rich cathode and flame-retardant effect. Then, the functional additives fluoroethylene carbonate (FEC) and dimethyl allylphosphate were introduced to triethyl phosphate electrolytes to stabilize the deposition of metallic lithium and ameliorate the electrode-electrolyte interface environment. Furthermore, the Li||Li and Li||NCM batteries show a favorable cycling, and a stable and phosphate-based SEI layer was in-situ constructed. The manuscript is well organized, and the results are clear and well supported by the data. The paper can be published on molecules after minor revision based on the following comments:

1) The authors reported in the manuscript that the designed electrolytes present an excellent anodic stability through lithium symmetrical batteries and SEM. The other result EDS should be supplemented in the manuscript.

2) In Figure 4, Li||NCM batteries with DMAPs exhibit much more stabilized charge-discharge. Please supply corresponding rates performance data.

3) Please add more discussions about organic phosphate as nonflammable electrolytes in the section of introduction. In addition, recent reports on the electrolytes of batteries, such as Small Methods 2021, 5, 2100441, Journal of Energy Chemistry 2021, 63, 270, Angew Chem Int Ed 2021, 60, 26837, and Science Bulletin 2022, 67, 1581, are suggested to be referred and cited appropriately.

4) Please provide a clearer description of the methods used for electrochemical tests such as LSV.

5) Some minor issues with typos and reference in the manuscript should be carefully corrected.

Author Response

For Reviewer #2:

This manuscript entitled “Ameliorating phosphonic-based nonflammable electrolytes towards safe and stable lithium metal batteries" developed a strategy to handle the interface compatibility and stability faced by nonflammable electrolytes via introducing functional additive dimethyl allyl-phosphate and fluoroethylene carbonate into triethyl phosphate electrolytes to stabilize the deposition of metallic lithium and accommodate the electrode-electrolyte interface. Dimethyl allyl-phosphonate shows superior compatibility with layered Ni-rich cathode and flame-retardant effect. Then, the functional additives fluoroethylene carbonate (FEC) and dimethyl allyl-phosphate were introduced to triethyl phosphate electrolytes to stabilize the deposition of metallic lithium and ameliorate the electrode-electrolyte interface environment. Furthermore. the Li||Li and Li||NCM batteries show a favorable cycling, and a stable and phosphate-based SEI layer was in-situ constructed.

Response: Thanks for your positive evaluation about our work.

  1. The author reported in the manuscript that the designed electrolytes present an excellent anodic stability through lithium symmetrical batteries and SEM. The other result EDS should be supplement in the manuscript.

Response 1: Thanks for the reviewer’s valuable advice. We will submit the supplementary information attachment about EDS spectra (see SI Figure 1). Please see the attachment.

  1. In Figure 4, Li||NCM batteries with DMAPs exhibit much more stabilized charge-discharge. Please supply corresponding rates performance data.

Response 2: Thanks for the reviewer’s suggestion. We will submit the supplementary information of corresponding rates performance figure. (see SI Figure 2). Please see the attachment. 

  1. Please add more discussion about organic phosphonate as nonflammable electrolytes in the section of introduction. In addition, recent reports on the electrolytes of batteries, such as Small Methods 2021,5,2100441, Journal of Energy Chemistry 2021, 63, 270, Angew Chem Int Ed 2021, 60, 26837, and Science Bulletin 2022, 67, 1581, are suggested to be referred and cited appropriately.

Response 3: Thanks for the valuable suggestions to improve the quality of our manuscript. We appreciate that the reviewer presents so important works to us that we have cited the mentioned papers as Refs. 51 to 54 and discusses these works in the introduction. And, highlighted in red these works in the revised manuscript (see Page 2, lines 13-18; see Page 12, lines 20-31).

  1. Please provide a clearer description of the methods used for electrochemical tests such as LSV.

Response 4: Thanks for the reviewer’s valuable suggestion. The decomposition voltage of test electrolyte is generally selected LSV test and used steel symmetric battery. The test starts with the open circuit voltage of the battery and goes all the way to the large range voltage we expect. Generally, 0.1mv/s is selected as the sweep speed. (see Page 4, lines 4-6).

  1. Some minor issues with typos and reference in the manuscript should be carefully corrected.

Response 4: Thanks for the reviewer’s valuable suggestion. We have seriously examined and corrected the unnecessary typos and English grammar errors in the revised manuscript. (see Page 5, line 14; see Page 3, line 4). 

Reviewer 3 Report

The manuscript authored by Sha Fu et al. reports on preparation of novel nonflammable electrolytes by introducing the functional additives fluoroethylene carbonate and dimethyl allylphosphate to triethyl phosphate electrolytes. This strategy allowed to increase the stability of the cathode and electrolyte interface and improve compatibility of electrolyte with metallic lithium anode. All this resulted in a superior cycling stability of the hybrid lithium batteries paired with phosphonic-based electrolytes after 200 and 450 cycles.

The article is well-designed and possesses enough scientific novelty to be published in “Molecules” after the authors address some questions:

1. Although the article is well organized, its English quality should be improved, especially in the Materials and Methods section. There are many errors that even free online editing services like Wordtune or Writefull would detect and correct.

2. The authors randomly use abbreviations for fluoroethylene carbonate and dimethylallylphosphate. Thus, they provide the abbreviation FEC for the first time in Introduction. However, they duplicate its description in the Results and Discussion (p. 2) and Conclusions (p. 8) sections. As for dimethylallylphosphate (by the way, it should be one-worded), its abbreviation DMAP appears for the first time only on page 3. Please check the whole manuscript.

3. The authors should clarify which of the electrolytes presented on Figs 1a and 1b are commercial and which are DMAP. I’d also recommend changing the description to Fig. 1.

4. It is important for the authors to explain carefully the difference in scale on SEM images of metallic Li anode surfaces in (b, c) commercial electrolytes and (d, e) DMAP electrolytes in Fig 4. Why can’t we see 1 μm images in Figs. 4d and 4e? Perhaps, lithium dendrites would be found in both cases if we zoomed up to 1 μm?

Author Response

For Reviewer #3:

The manuscript authored by Sha Fu et al. reports on preparation of novel nonflammable electrolytes by introducing the functional additives fluoroethylene carbonate and dimethyl allylphosphate to triethyl phosphate electrolytes. This strategy allowed to increase the stability of the cathode and electrolyte interface and improve compatibility of electrolyte with metallic lithium anode. All this resulted in a superior cycling stability of the hybrid lithium batteries paired with phosphonic-based electrolytes after 200 and 450 cycles

The article is well-designed and possesses enough scientific novelty to be published in Molecules” after the authors address some questions:

Response: Thanks for your positive evaluation about our work.

  1. Although the article is well organized, its English quality should be improved, especially in the Materials and Methods section. There are many errors that even free online editing services like Wordtune or Writefull would detect and correct.

Response 1: Thanks for the comments. We have corrected some misspellings in the names of substances in the revised manuscript. (see Page 7, line 11-13 and line 16; see page 8, line 7-8).

     2. The authors randomly use abbreviations for fluoroethylene carbonate and dimethyl allyl-phosphate. Thus, they provide the abbreviation FEC for the first time in Introduction. However, they duplicate its description in the Results and Discussion (p. 2) and Conclusions (p. 8) section. As for dimethyl allyl-phosphate (by the way, it should be one-worded), its abbreviation DMAP appears for the first time only on page 3. Please check the whole manuscript.

Response 2: Thanks for the suggestion. We have carefully examined and corrected errors in the revised manuscript (see Page 2, line 20-23, line 46-48; see Page 3, line 2).

     3 The authors should clarify which of the electrolytes presented on Figs 1a and 1b are commercial and which are DMAP. I'd also recommend changing the description to Fig. 1.

Response 3: Thank you for your kind comments on our manuscript, and we have carefully corrected related description in the revised manuscript according to the valuable suggestion (see Page 2, lines 48-50).

     4 It is important for the authors to explain carefully the difference in scale on SEM images of metallic Li anode surfaces in (b, c) commercial electrolytes and (d, e) DMAP electrolytes in Fig 4. Why can't we see 1 um images in Figs.4d and 4e? Perhaps, lithium dendrites would be found in both cases if we zoomed up to 1 um?

Response 4: Thank you for your kind comments on our manuscript. We think that the magnification of 1μm on the surface (b, c) of the lithium electrode is helpful to observe the dendrites or other adverse phenomena occur in commercial electrolytes. And, the cross section (d, e) is to see how thick the SEI layer in DMAP electrolytes and whether it is peeling off on the lithium foil. In the test of a single cell, we have observed the similar deposits of lithium patterns magnified to 1μm in DMAP electrolytes. Mainly to highlight the DMAP electrolyte formed SEI density, shape and other properties are better in Fig 4.

Reviewer 4 Report

The manuscript is well-designed and well-written. I have several minor suggestion:

1. Please add experimental details for the experiment at different temperature (Figure 2b).

2. Is Figure 3d a cross-sectional image? What is 90.9 um for? 

3. For the fitted peaks in XPS, please add references to support the peak fitting.

Author Response

For Reviewer #4:

The manuscript is well-designed and well-written. I have several minor suggestion:

Response: Thanks for your positive evaluation about our work.

  1. Please add experimental details for the experiment at different temperature (Figure 2b).

Response 1: Thanks for the comments. We have added some experimental steps in the revised manuscript. (see Page 4, line 1-3).

     2. Is Figure 3d a cross-sectional image? What is 90.9 μm for?

Response 2: Thanks for the suggestion. Figure 3d is a cross-sectional image. 90.9μm is SEI formed when lithium is uniformly deposited on the electrode surface. (see Page 4, line 9-10).

     3 For the fitted peaks in XPS, please add references to support the peak fitting.

Response 3: Thank you for your kind comments on our manuscript, and we fitted peaks in XPS according ref 34-35, and corresponding information has been added to the revised manuscript. (see Page 7, line 8).

Round 2

Reviewer 3 Report

All comments were addressed by the authors. The manuscript can be accepted for publication in Molecules in present form.